# The Role of Neutrophil Extracellular Networks in Cardiovascular Pathology

**DOI:** 10.3390/cells14191562

**Published:** 2025-10-08

**Authors:** Zofia Szymańska, Antoni Staniewski, Michał Karpiński, Katarzyna Zalewska, Oliwia Kalus, Zofia Gramala, Joanna Maćkowiak, Sebastian Mertowski, Krzysztof J. Filipiak, Mansur Rahnama-Hezavah, Ewelina Grywalska, Tomasz Urbanowicz

**Affiliations:** 1Cardiology Research Student Group, Poznan University of Medical Sciences, 61-701 Poznan, Poland; 2Department of Experimental Immunology, Medical University of Lublin, 4a Chodźki Street, 20-093 Lublin, Poland; 3The Centre of Postgraduate Medical Education, 99/103 Marymoncka Street, 01-813 Warsaw, Poland; 4Department of Dental Surgery, Medical University of Lublin, 6 Chodźki Street, 20-093 Lublin, Poland; 5Cardiac Surgery and Transplantology Department, Poznan University of Medical Sciences, ½ Długa, 61-848 Poznan, Poland

**Keywords:** neutrophil extracellular traps, atherosclerosis, immunothrombosis, coronary artery disease, endothelial dysfunction, oxidative stress, platelets, biomarkers, reperfusion injury, pharmacotherapy

## Abstract

Cardiovascular diseases (CVDs) are increasingly being defined not only in terms of metabolic or purely vascular disorders, but also as complex immunometabolic disorders. One of the most groundbreaking discoveries in recent years is the role of neutrophil extracellular networks (NETs/NENs) as a key link between chronic vascular wall inflammation and thrombotic processes. In this article, we present a synthetic overview of the latest data on the biology of NETs/NENs and their impact on the development of atherosclerosis, endothelial dysfunction, and the mechanisms of immunothrombosis. We highlight how these structures contribute to the weakening of atherosclerotic plaque stability, impaired endothelial barrier integrity, platelet activation, and the initiation of the coagulation cascade. We also discuss the modulating role of classic risk factors such as hypertension, dyslipidemia, and exposure to tobacco smoke, which may increase the formation or hinder the elimination of NETs/NENs. We also focus on the practical application of this knowledge: we present biomarkers associated with the presence of NETs/NENs (cfDNA, MPO–DNA complexes, CitH3, NE), which may be useful in diagnostics and risk stratification, and we discuss innovative therapeutic strategies. In addition to classic methods for indirectly inhibiting NET/NEN formation (antiplatelet, anti-inflammatory, and immunometabolic agents), we present experimental approaches aimed at their neutralization and removal (e.g., DNase I, elastase, and myeloperoxidase inhibitors). We pay particular attention to the context of cardiac and cardiac surgical procedures (Percutaneous Coronary Intervention-PCI, coronary artery bypass grafting-CABG), where rapid NET/NEN bursts can increase the risk of acute thrombotic complications. The overall evidence indicates that NETs/NENs represent an innovative and promising research and therapeutic target, allowing us to view cardiovascular diseases in a new light—as a dynamic interaction of inflammatory, atherosclerotic, and thrombotic processes. This opens up new possibilities in diagnostics, combination treatment and personalisation of therapy, although further research and standardization of detection methods remain necessary.

## 1. Introduction

Cardiovascular diseases (CVDs) remain the leading cause of death worldwide, accounting for millions of deaths annually and placing a considerable burden on healthcare systems. In 2021, heart disease and stroke were responsible for 20.5 million deaths, representing approximately one-third of all deaths globally [1]. Coronary artery disease (CAD) is particularly concerning, causing ~9 million deaths annually and affecting over 250 million people [2]. Notably, after a period of decline in the age-adjusted CVD mortality rate between 2010 and 2019 (−8.9%), a reversal was observed between 2019 and 2022 (+9.3%; 454.5 deaths/100,000), and in 2020, heart disease and stroke surpassed cancer and chronic lower respiratory diseases combined as the leading cause of death [3,4].

Among CVDs, atherosclerotic diseases predominate, primarily ischemic heart disease (IHD), also known as CAD or atherosclerotic cardiovascular disease (ACD), which clinically most often manifests as myocardial infarction and ischemic cardiomyopathy [5]. The atherosclerosis paradigm now extends beyond the classic “lipid hypothesis”: the immune system plays a crucial role in the initiation and progression of changes. Chronic inflammation of the vascular wall—driven by monocytes/macrophages, neutrophils, and cytokine cascades—deepens endothelial dysfunction, increases oxidative stress, and prothrombotic activity, creating an environment conducive to plaque formation and destabilization [6]. The immunological approach encompasses both the mechanisms of innate immunity (including granulocyte recruitment and complement activation) and adaptive immunity (antigen presentation, activation, and modulation of lymphocyte function), as well as the concept of “trained immunity,” which perpetuates the pro-inflammatory phenotype of myeloid cells [7,8,9].

In this context, the role of neutrophil extracellular traps (NETs)—fibrous structures composed of decondensed chromatin, histones, and granule proteins (including MPO, NE, and cathepsin G), released by activated neutrophils during the process of netosis—is of particular interest. NETs constitute an effective barrier against microorganisms, but in the setting of sterile vascular inflammation, they can act as damage-associated molecular patterns (DAMPs), sustaining and escalating the inflammatory response [10,11]. Experimental and clinical data, as they promote LDL oxidation. This suggests that NETs are related to the pathogenesis of atherosclerosis and thrombosis, as they contribute to the formation of foam cells, degradation of the extracellular matrix, and thinning of the fibrous plaque cap. They also provide a scaffold for platelet adhesion and activation, bind coagulation factors, and enhance thrombin generation. Consequently, they participate in both the initiation and progression of atherosclerotic lesions, as well as in their destabilization and ischemic complications [12,13,14].

The importance of NETs extends beyond the coronary artery. Their presence and activity have been demonstrated in carotid plaques and thrombi, as well as in peripheral artery disease, correlating with inflammatory markers and disease severity [15,16]. Furthermore, CVD risk factors—such as hypertension, dyslipidemia, and tobacco smoke exposure—modulate the formation and/or degradation of NETs, creating positive feedback loops between environmental and metabolic stress and neutrophil inflammatory activity [17,18]. These observations paint a coherent yet complex picture in which NETs constitute a central node in the pathophysiological network leading to cardiovascular events. The practical implications of this perspective are twofold. First, NET markers (e.g., extracellular DNA, citrullinated histones, MPO-DNA) may serve as indicators of disease activity and risk of complications. However, they require standardization of assay methods and clinical validation. Second, the possibility of targeted therapeutic intervention arises: from indirect inhibition of NETosis by limiting platelet activation and oxidative stress, to modification of inflammatory axes and—potentially—direct influence on NETs formation/degradation. Simultaneously, surgical procedures (restoring perfusion) may reduce the stimuli triggering NETosis, which is reflected in the dynamics of biomarkers after revascularization [3,4,10,11,12,13,14,15,16,17,18].

This narrative review synthesizes the current understanding of NETs in the pathogenesis of CVDs, from the basics of neutrophil biology and the mechanisms of NETosis, through their impact on atherosclerosis, endothelial dysfunction, and thrombosis, to data on coronary artery disease, carotid artery pathology, and peripheral artery disease. We also discuss translational implications and potential therapeutic directions, and identify research gaps relevant to the design of future studies. In light of the persistent global burden of CVD, a deeper understanding of the role of NETs may lead to more effective risk stratification and more precise immunomodulatory treatment strategies.

## 2. Immunological Basis of Atherosclerosis Pathogenesis: From Lipoprotein Retention to Immunothrombosis

Atherosclerosis is a chronic lipid-inflammatory disease of the arterial wall that initiates in sites of impaired hemodynamics (low/oscillatory shear), where favorable permeability and retention conditions lead to the deposition of apoB-dependent lipoproteins in the intima. LDL (low-density lipoproteins) bound to intimal proteoglycans undergo oxidative and enzymatic modifications, generating neoepitopes recognized by pattern receptors, which activate endothelial cells and trigger Nuclear factor kappa-light-chain-enhancer of activated B cells (NF-κB)-dependent transcription. This increases barrier permeability and the expression of adhesion molecules (vascular cell adhesion protein 1- (VCAM-1), Intercellular Adhesion Molecule 1 (ICAM-1), selectins) and chemokines (e.g., CCL2, CXCL1, CXCL8), resulting in targeted recruitment of monocytes and neutrophils to the site of the developing lesion [6,19,20]. In this way, the early lipid signal is translated into a sustained immune response in the vascular wall.

Then, after crossing the endothelium (platelet endothelial cell adhesion molecule-1 (PECAM-1)), monocytes differentiate into macrophages and intensively phagocytize modified lipoproteins via scavenger receptors (CD36, Scavenger Receptor A1 (SR-A1), Lectin-like oxidized low-density lipoprotein receptor-1 (LOX-1)). This occurs due to the lack of cholesterol feedback, leading to the accumulation of cholesteryl esters and the formation of foam cells. Metabolically reprogrammed macrophages adopt a pro-inflammatory phenotype and secrete IL-1β, tumor necrosis factor (TNF-α), IL-6, and matrix metalloproteinases and cathepsins, which degrade the scaffolding components of the vessel wall and weaken the fibrous cap [6,21,22]. Simultaneously, neutrophils are recruited in this process, which amplifies the inflammatory component by supplying myeloperoxidase and elastase, generating ROS. Moreover, in response to PAMP/DAMP signals and interactions with platelets, they also release NETs, which act as injury templates, damage the endothelium, and provide a prothrombotic scaffold [21,22]. Furthermore, the complement system is also involved in this process, via anaphylatoxins C3a/C5a and the membrane attack complex. It enhances leukocyte chemotaxis and activation, and the deposition of complement components on lipoproteins and vessel wall cells perpetuates the inflammatory reaction [23].

As lipid burden and oxidative stress increase, innate immune sensors, including the nucleotide-binding domain, leucine-rich-containing family, and Pyrin domain-containing 3 (NLRP3) inflammasome, are activated in macrophages. Crystalline cholesterol and oxidized lipids act as DAMPs, initiating IL-1β/IL-18 maturation and amplifying the positive feedback loop, resulting in increased leukocyte counts, higher cytokine concentrations, and further endothelial dysfunction and vessel wall remodeling [6,22]. Simultaneously, NO bioavailability decreases (eNOS uncoupling), which deepens endothelial dysfunction and promotes vascular constriction [24]. A key step in the transition toward chronic instability is efferocytosis failure—impaired removal of dying cells and cellular debris due to factors such as proteolytic cleavage of the Myelo–Epithelial–Reproductive Tyrosine Kinase (MerTK) receptor and a deficiency in bridging proteins. Retention of this cellular debris promotes the enlargement of the necrotic core and perpetuates inflammation, simultaneously limiting the ability of vascular smooth muscle cells (VSMCs) to rebuild their collagen cap [25,26].

Long-term exposure to pro-atherogenic factors (oxLDL axis, but also hyperglycemia and signals controlling myelopoiesis) induces “trained immunity.” Epigenetic reprogramming of myeloid precursors (e.g., H3K4me3 markers of inflammatory gene promoters) and a metabolic switch toward glycolysis result in the perpetuation of the pro-inflammatory phenotype of macrophages and neutrophils, which persists even as stimulus exposure abates, leaving the vessel wall in a state of low-grade, chronic inflammation [9]. As the lesion itself matures, a component of adaptive immunity also increases. Dendritic cells capture lipid and protein antigens associated with oxLDL/apoB, mature, and present them in the context of major histocompatibility complex (MHC) to T lymphocytes both within the plaque and in perivascular nodes. The dominant Th1 (IFN-γ) and Th17 (IL-17) responses are pro-atherogenic, meaning that interferon gamma (IFN-γ) activates macrophages and smooth muscle cells, increases matrix metalloproteinase (MMP) expression, and attenuates cap fibrosis. At the same time, IL-17 promotes the recruitment of neutrophils and the production of chemokines [22]. Immune checkpoints present in plaques (e.g., PD-1 (Programmed cell death-1; CD279); PD-L1 (Programmed death-ligand 1; CD274); CTLA-4 (Cytotoxic T-lymphocyte-associated protein 4; CD152)) act as physiological inhibitors of this response—their blockade increases inflammation and lesion instability [27]. In turn, Treg lymphocytes (producing IL-10 and TGF-β) have a protective effect by promoting efferocytosis and suppressing macrophage activation; however, their number and function are sometimes reduced in plaques. In parallel, B lymphocyte populations modulate the course of the process: B2 cells produce IgG against lipid neoepitopes and B-cell activating factor (BAFF), which is associated with the severity of inflammation. At the same time, natural IgM (mainly from B1) against oxLDL epitopes facilitates clearance and may have a protective effect [22].

As indicated in the literature, smooth muscle remodeling is also and crucial component of this immune axis. VSMCs migrate to the intima and undergo phenotypic reprogramming, producing collagen in the fibrous cap. However, under the influence of cytokines and ROS, they limit collagen biosynthesis and increase protease expression, thinning the cap and increasing susceptibility to erosion or rupture. Under normal conditions of inflammation resolution, specialized pro-resolving mediators (SPMs), including lipoxins, resolvins, protectins, and maresins, restore homeostasis by enhancing efferocytosis and suppressing cytokine expression. However, their deficiency is observed in atherosclerotic plaques, which perpetuates incomplete resolution and chronic inflammation [28,29]. It should be noted that these interactions result in the formation of a lesion susceptible to thrombotic complications. Matrix degradation and cap thinning facilitate erosion and rupture, exposing thrombogenic material. At the same time, NETs and innate immune response components link inflammation with immunothrombosis, as DNA-histone networks bind von Willebrand factor (vWF), activate factor XII, recruit and activate platelets, and improve thrombin generation. This translates into clinical cardiovascular events (from acute coronary syndromes to cerebrovascular events) and provides a basis for consideration of risk biomarkers and therapeutic interventions aimed at modifying the inflammatory response (Figure 1) [19,21,22,28].

## 3. Neutrophils and the Mechanisms of Formation of Neutrophil Extracellular Networks (NENs)

Neutrophils produced in the bone marrow during granulopoiesis constitute ~2/3 of peripheral blood leukocytes and represent the most rapidly mobilized effector cells of innate immunity [30,31,32]. After recruitment to inflammatory sites through rolling, integrin-dependent adhesion, and diapedesis, neutrophils initiate a broad repertoire of effector functions, including phagocytosis, production and release of reactive oxygen species (ROS), degranulation with the emission of antimicrobial proteins, secretion of cytokines and chemokines, and—of particular importance for vascular pathophysiology—the formation of NETs (Figure 2) [32,33,34,35].

NETs are fibrous structures composed of decondensed chromatin (predominantly nuclear DNA, though in some contexts also mitochondrial DNA) decorated with histones and granule proteins. Among their key components are calprotectin (S100A8/A9), myeloperoxidase (MPO), neutrophil elastase (NE), cathepsin G, and histones H2A/H2B/H3/H4, which not only provide the architectural scaffold but also confer potent antimicrobial activity [36,37,38]. This structural organization functions as a physical “glue” that immobilizes pathogens while simultaneously serving as a localized hub for antimicrobial effectors.

Depending on the nature of the stimulus and the signaling pathway, three main mechanisms can be distinguished:Suicidal NETosis—a form of programmed cell death alternative to apoptosis, in which, within a few hours, massive nuclear envelope disintegration, chromatin decondensation, and cell lysis occur, with the release of DNA lattice. In this variant, the key source of ROS is NADPH oxidase (NOX2) [39,40,41].Vital NETosis—rapid (≤~1 h) release of chromatin fragments in vesicles or mtDNA ejection without loss of plasma membrane integrity; the neutrophil retains the ability to migrate and phagocytose. This variant is NOX2-independent and relies on mitochondrial-derived ROS [39,40,41,42].Additionally, mitochondrial NETosis (mtNETosis) has been described—a usually nonlytic variant in which oxidized mtDNA is exocytosed in vesicles; the process is essentially NOX2-independent, dependent on mROS and Ca^2+^ influx/mPTP opening, and often overlaps with the viable form [43,44,45] (Figure 3).

Regardless of the variant, key steps include the following:ROS generation (NOX2 vs. mitochondria),NE and MPO translocation to the nucleus,histone modifications—including PAD4-catalyzed citrullination—leading to chromatin loosening,cytoskeletal rearrangement and disruption of the nuclear envelope integrity,extrusion of DNA fibers coated with granule proteins into the extracellular space [39,40,41].

In parallel, kinase pathways (e.g., PKC-dependent in PMA models) and calcium-dependent pathways are activated, which modulate the intensity and dynamics of the response.

Physiologically, NETs support pathogen elimination by immobilizing microbes, enhancing local protease/oxidase activity, and providing substrates for phagocytes. In sterile inflammation, however, these same features become pathogenic: NET components act as DAMPs, amplify inflammatory signaling, damage tissues, form prothrombotic scaffolds, and contribute to the formation and destabilization of atherosclerotic plaques [46,47]. The clinical outcome depends on the balance between NET generation and clearance, and its disruption sustains inflammation, endothelial dysfunction, and thrombosis, particularly in CVDs. NETs have been identified in both atherosclerotic plaques and arterial thrombi, linking lesion initiation and progression to thrombotic complications [11,48,49]. Mechanistically, histones, elastase, and MPO disrupt endothelial integrity, promote adhesion molecule expression, oxidize LDL, and degrade extracellular matrix components, thereby driving foam cell formation, cap thinning, and plaque instability. The DNA–protein framework also provides a scaffold for platelet activation and thrombin generation, directly enhancing arterial thrombosis [50,51,52]. Collectively, excessive or persistent NET activity bridges the inflammatory–lipid axis with prothrombotic processes and vessel wall remodeling, underpinning their central role in atherosclerosis and immunothrombosis [48,50,51].

Although the term NET is most widely used, the concept of neutrophil extracellular networks (NENs) has emerged as a broader definition, emphasizing the three-dimensional and interconnected network character of these structures. While every NET can be considered a form of NEN, the latter underlines the spatial architecture and multifunctional interactions of neutrophil-derived extracellular fibers with the surrounding immune and stromal microenvironment. Thus, NETs represent a mechanistically defined subset of the broader NEN phenomenon (Table 1).

Immunoinflammation is a currently recognized mechanism linking chronic inflammation with dysregulation of the immune and metabolic response. In this process, neutrophils, monocytes, and lymphocytes are in constant dialog with endothelial cells and extracellular matrix components, leading to the amplification of inflammatory signals and the perpetuation of a state of low-grade, chronic inflammation. Excessive neutrophil activation results in increased ROS production, cytokine secretion, and the formation of extracellular structures (NETs/NENs), which act as endogenous DAMPs. This leads to further leukocyte recruitment, endothelial damage, and platelet activation, linking immunoinflammation with thrombotic processes and the development of atherosclerosis [7,54,55].

In this context, the neutrophil-to-lymphocyte ratio (NLR) is particularly valuable, as it reflects the balance between the pro-inflammatory activity of neutrophils and the regulatory role of lymphocytes [56]. Elevated NLR values indicate a relative predominance of neutrophil-driven pro-inflammatory activity over lymphocyte-mediated adaptive responses, thereby capturing the essence of immunoinflammation. Clinical data demonstrate that NLR is a strong predictor of atherosclerotic carotid plaques in older adults, independent of traditional cardiovascular risk factors, underscoring the contribution of chronic low-grade inflammation to plaque formation and progression. Mechanistically, higher NLR may reflect enhanced neutrophil activation and a greater propensity for NET/NEN release, which in turn act as DAMPs, damage the endothelium, promote LDL oxidation and foam cell formation, and provide a prothrombotic scaffold. This positions NLR not merely as a general marker of systemic inflammation, but as an integrative index linking innate immune dysregulation, vascular remodeling, and thrombo-inflammatory processes characteristic of atherosclerosis and its complications [56]. In the study, Assessing Humoral Immuno-Inflammatory Pathways Associated with Respiratory Failure, the authors analyzed the dynamics of immuno-inflammatory markers, such as neutrophil and lymphocyte counts, and CRP levels, in COVID-19 patients during hospitalization to identify hyperinflammatory states and predict respiratory deterioration [57]. They noted that neutrophilia, lymphopenia, and elevated CRP levels correlate with advanced inflammation and worse clinical outcomes. In the context of immunoinflammation, such observations are crucial—they demonstrate how pro-inflammatory neutrophil activity (with a reduced lymphocyte response) can outweigh regulatory mechanisms, perpetuating inflammation and promoting tissue damage. Similarly, in the concept of CVDs, an increased NLR can reflect a predominance of immuno-inflammatory activity, potentially increasing the risk of developing structures such as NETs/NENs and engaging immunothrombotic mechanisms.

### 3.1. The oxLDL–NET Axis in the Initiation, Progression, and Destabilization of Atherosclerotic Plaque

Oxidized LDL lipoproteins (oxLDL) constitute a key impulse initiating and amplifying the atherosclerotic cascade and simultaneously promote NETosis. OxLDL has been shown to enhance NET release (e.g., in models using PMA, a PKC activator) [58,59], which results from their DAMP properties—oxidized phospholipids and lysophosphatidylcholine (lysoPC) activate pattern receptors and neutrophil inflammatory programs [60]. Additionally, oxLDL drives a positive feedback loop: monocytes phagocytize oxLDL, differentiate into oxLDL-fed macrophages, and secrete miR-146a-containing exosomes, which, when taken up by neutrophils, inhibit SOD2 expression, increase ROS production, and induce NET formation [60]. In this way, the lipid signal is translated into a sustained activation of the innate immune response by neutrophils. NET products enhance pathogenesis at several levels. MPO enhances ROS production, LDL oxidation, and endothelial dysfunction by “consuming” NO [58]; NE proteolyzes LDL components (apoAII, apoBII), increasing their uptake by macrophages; cathepsin G degrades apoB-100, promotes LDL fusion, and increases their binding to aortic proteoglycans and atherosclerotic foci. This creates a vicious cycle: oxLDL → NETs → further LDL modification and oxLDL growth → plaque progression. Experimental data indicate that the pro-atherogenic effect of NETs appears early. In mice fed a high-fat diet, NET structures were detectable after only 3 weeks, and their number increased over time [61,62]. Notably, the deletion of PAD4—an enzyme essential for NETosis—in the bone marrow lineage significantly reduced the atherosclerotic burden after 10 weeks, confirming the causative role of NETs [52,63]. The propagation mechanism also involves macrophage activation by NET components and the secondary induction of IL-1β, CCL2, IL-8, and IL-6, which sustain leukocyte recruitment and vessel wall remodeling [64,65,66,67]. NETs also participate in the transition to plaque instability. Their presence was observed in complex foci (erosion/rupture), whereas intact plaques showed minimal NET signal [40]. In cell models, the combination of NET-containing medium with oxLDL increased MMP-1 expression in aortic endothelial cells (HAECs) [60]. Increased activity of MMP-1—an interstitial collagenase—promotes the degradation of collagens I and III and the erosion of the fibrous cap [67]. Specific NET components also contribute to destabilization: MPO promotes cap thinning [68,69], and histone H4 can bind to vascular smooth muscle cells, inducing their lysis and weakening the mechanical support of the plaque [70,71]. Overall, the oxLDL–NET axis integrates lipid, inflammatory, and proteolytic signals, driving the initiation, progression, and rupture of atherosclerotic lesions [64,65,66,67,68,69,70,71].

### 3.2. NETs as Mediators of Endothelial Dysfunction and Immunothrombosis

Literature studies suggest that the role of NETs extends beyond their involvement in atherosclerosis, as they directly drive immunothrombosis by simultaneously damaging the endothelial barrier and initiating coagulation cascades. A key element of this axis is the histone components of NETs. Citrullinated histone H3 (CitH3) increases leakage in the microcirculation and reduces transendothelial electrical resistance (TER), reflecting the loss of integrity and tightness of intercellular junctions [72]. Mechanistically, CitH3 disrupts the architecture of the actin cytoskeleton by polymerizing G-actin and binding F-actin [73] and by “thinning” VE-cadherin in adherens zones, creating gaps between endothelial cells [74]. In parallel, selected NET components—particularly cathepsin G and IL-1α—activate endothelial cells (ECs), inducing a pro-inflammatory and procoagulant phenotype, as well as superficial erosion, which predisposes to arterial thrombosis [36]. Once the barrier is breached, the fibrous DNA-histone network acts as a scaffold for hemostasis, facilitating platelet adhesion, activation, and aggregation. Additionally, histones can directly activate platelets in a P-selectin- or TLR2/4-dependent manner [75,76,77,78]. NETs bind vWF, fibronectin, and fibrinogen, stabilizing newly formed thrombi, including erythrocyte-rich thrombi [79,80]. At the molecular scale, NETs bind and activate factor XII, initiating the contact pathway of coagulation [81]. They also express functional tissue factor (TF), which accelerates thrombin generation and secondary platelet activation, thereby closing a positive feedback loop [81]. Platelets themselves, in turn, enhance NETosis by interacting with neutrophils via PSGL-1–P-selectin and releasing CXCL4 (PF4), thromboxane A_2_, and vWF, thereby intensifying NET formation in situ within the thrombus [82,83,84]. An essential regulatory node is the vWF–ADAMTS13 axis. Inflammation reduces the availability of ADAMTS13, an enzyme that physiologically cleaves ultralarge vWF multimers [85]. Additionally, PAD4 is released with NETs, citrullinates, and impairs ADAMTS13, which promotes the persistence of ultralarge vWF strands and the formation of highly thrombogenic platelet “strings” [86]. vWF itself enhances the loop, promoting platelet-induced NET formation through interactions of GPIbα (platelets) and αMβ2 (neutrophils) [87,88]. NETs also weaken anticoagulant brakes. NE and cathepsin G, as well as extracellular nucleosomes, proteolyze TFPI—a significant inhibitor of the tissue factor pathway—pave the way for both TF-dependent and FXII-dependent activation of coagulation [89]. Histones further disrupt the formation of activated protein C (APC) by interfering with the interaction between protein C and thrombomodulin. Since APC digests histones, APC inhibition paradoxically enhances histone toxicity and thrombotic phenomena [90]. Furthermore, histone H4 can promote prothrombin autoactivation, enhancing local thrombin production independently of classical initiators [91] (Figure 4). These processes partially explain how NETs link inflammatory damage to the vascular barrier with coagulation amplification and thrombus stabilization, thereby creating a self-sustaining immunothrombotic loop in cardiovascular diseases [72,73,74,75,76,77,78,79,80,81,82,83,84,85,86,87,88,89,90,91,92].

## 4. NETs as a Bridge Between Pathophysiological Conditions and the Cardiovascular Phenotype

From a physiological perspective, NETs act as a rapid “threat limiter” and a catalyst for pathogen clearance. Under conditions of chronic metabolic, oxidative, or hemodynamic stress, the balance between their formation and clearance is disrupted, shifting the NET’s function from protective to pathogenic. This leads to an amplification of the inflammatory–prothrombotic axis, leading to endothelial dysfunction, lipoprotein modification, platelet activation and coagulation, and disruption of anticoagulant and resolution mechanisms [47,93]. Based on available literature data, Table 2 presents the relationships between NETs and selected pathophysiological conditions and CVDs, demonstrating how systemic factors modulate NETs activity and pave the way for vascular complications.

Increased levels of NET markers are observed in patients with hypertension, and PAD4-deficient models indicate that reduced NET formation capacity mitigates blood pressure elevation and improves vascular relaxation [94,95]. These data suggest that NET components, including histones, activate the endothelium and impair vascular function, and promote smooth muscle cell proliferation via the Akt/CDKN1b/TK1 axis, increasing vascular resistance [96]. Clinically, NET formation is enhanced by classical prohypertensive factors, such as angiotensin II or isolevuglandins, which complete the positive feedback loop between blood pressure regulation and immunothrombosis [97]. Dyslipidemia contributes to enhanced NET formation, triggered by oxLDL and oxidized phospholipids, while simultaneously impairing their clearance through reduced deoxyribonuclease activity, thereby facilitating NET accumulation within the vascular compartment [98,99].In parallel, cholesterol accumulating in macrophages activates the NLRP3 inflammasome and IL-1β secretion, which secondarily induces NETosis, enhancing vascular wall inflammation [99]. This lipid–inflammatory axis is further modulated by HDL, which inhibits NETs formation, suggesting the potential for risk modification through metabolic interventions [59].

Research indicates that exposure to tobacco smoke intensifies the formation of NETs, mainly through an increase in ROS and the activation of the cGAS and TLR9 pathways by DNA perceived as DAMPs, which enhances NF-κB transcription and chronic inflammation (important, among other factors, in the pathogenesis of COPD) [100,101]. Long-term exposure to this stimulus leads to neutrophil reprogramming (resistance to ferroptosis, predominance of “viable” NETosis) and a simultaneous decrease in the activity of DNases that degrade the network, which cumulates tissue exposure to NET components [103].

In clinical practice, multiple traces of NET activity have been identified across the spectrum of CVD. In CAD, granulocytes and neutrophil elastase are consistently detected within atherosclerotic plaques, and a high burden of NETs in coronary thrombi correlates with more severe left ventricular contractile dysfunction and worse clinical outcomes [99,100,101,102,103,104,105,106]. Importantly, NETs not only participate in thrombus formation but also hinder thrombolysis, which has direct therapeutic implications for the efficacy of fibrinolytic therapy in acute coronary syndromes. Furthermore, beyond their prothrombotic role, experimental evidence suggests that NETs may exacerbate myocardial injury through the induction of autophagy and apoptosis in cardiomyocytes, although these findings still require robust clinical validation [102,107].

Observations from carotid artery disease provide additional insight, as NETs have been shown to promote intraplaque neovascularization—a process that increases plaque vulnerability and predisposes to rupture. This is accompanied by a shift in hemostatic balance towards coagulation at the expense of fibrinolysis, amplifying the thrombogenic potential of the lesion [103,104,105,106,107,108,109,110]. Clinically, this mechanistic insight helps explain why patients with a higher intraplaque inflammatory load are more prone to ischemic cerebrovascular events, and it underscores the need for risk stratification strategies incorporating NET-related biomarkers.

In PAD, elevated levels of NET-associated markers—including MPO, citrullinated histone H3 (CitH3), and circulating extracellular DNA—have been documented, along with the presence of extracellular DNA within thrombotic material itself [106,107,108,109,110,111,112,113]. Importantly, easily measurable clinical indices such as the neutrophil-to-lymphocyte ratio and whole blood viscosity demonstrate positive correlations with disease severity, suggesting that NETs may serve as a unifying determinant of both vascular inflammation and hypercoagulability. From a translational perspective, these findings provide a rationale for incorporating NET signatures into diagnostic and prognostic panels, and they highlight potential therapeutic targets aimed at limiting NET formation or accelerating their clearance.

Taken together, the clinical evidence positions NETs as biomarkers and effectors of disease severity, plaque instability, and poor outcomes across CAD, carotid artery disease, and PAD. Their dual role in amplifying vascular inflammation and creating a prothrombotic milieu highlights them as promising but still underexplored therapeutic targets in contemporary cardiovascular medicine.

## 5. NET and Response to Applied Therapies

As we have described, NETs constitute a key link between chronic inflammation, atherosclerosis, and thrombosis. At the systemic level, NETs integrate signals from platelets, endothelium, macrophages, and the coagulation cascade. At the molecular level, they integrate the cargo of proteases (NE, cathepsin G), oxidases (MPO), and histones with the DNA scaffold. Therefore, therapeutic interventions can be designed along four complementary axes, which together can form a “stimulus-to-effects” framework (Figure 5) [114,115,116,117,118].

Dividing interventions into the upstream–midstream–downstream axes and the IR context may be clinically useful because it allows for precise mapping of therapies to specific target sites (e.g., colchicine → upstream/priming, aspirin → upstream/platelet–neutrophil interactions, metformin → upstream/ROS/metabolism, clearance-enhancing strategies → downstream), facilitates the design of combination therapies that simultaneously limit new NET bursts and accelerate the removal of existing NETs, and also organizes pharmacodynamic monitoring—different biomarkers are relevant for the upstream axis (e.g., IL-1β, CXCL8, TXB_2_) and others for the midstream/downstream axis (e.g., CitH3, MPO–DNA complexes, cfDNA). Moreover, such a division favors thinking about safety, because in the upstream axis, typical limitations are the risk of bleeding (with platelet modification) and immunosuppression (with potent anti-inflammatory drugs), while in the downstream axis, the selectivity of action against clots and the possibility of undesirable interactions with fibrinolysis become crucial.

### 5.1. Pharmacotherapy

The starting point is to treat NETs as a common denominator for inflammation, atherosclerosis, and thrombosis [119]. Therefore, treatment should be layered: upstream—limiting the signals that trigger NETs in the first place; midstream—reducing the toxicity of already formed networks; downstream—dismantling them more quickly and facilitating the lysis of NET-rich clots. In practice, the most robust clinical data concern anti-inflammatory, antiplatelet, and immunometabolic drugs, many of which act on multiple axes simultaneously (Table 3).

### 5.2. Procedural Interventions and the Ischemia–Reperfusion Context

Myocardial revascularization (primarily Percutaneous Coronary Intervention- PCI and/or CABG) modifies the inflammatory environment in two opposing ways. On the one hand, it removes the chronic ischemic stimulus, reducing DAMP generation and the “sustaining” of NETosis over time; on the other, reperfusion itself induces a short, intense “burst” of NETosis driven by a sudden increase in ROS, platelet activation, transient damage to the endothelial glycocalyx, and rapid leukocyte adhesion to the vessel wall. Clinical observations in patients after PCI for STEMI and stable angina indicate an increase in NET markers approximately 3 h after the procedure, followed by a marked decrease within 2 weeks [147]; NET biomarkers remain lower than baseline values at 4 months [143]. This model is consistent with the mechanics of ischemia–reperfusion: restoration of flow exposes the endothelium to activated platelets (P-selectin–PSGL-1), triggers the ROS cascade (NOX2/mitochondria), increases barrier permeability, and facilitates contact of neutrophils with DAMP stimuli; after resolution of hypoxia and normalization of hemodynamics, sterile inflammatory stimuli diminish and omental clearance (DNase/efferocytosis) becomes dominant [147,148].

In PCI, the presence of a thrombus rich in NETs and lipid–protein material increases the risk of distal embolization and no-reflow phenomena, as DNA-histone fibers create a scaffold resistant to purely fibrinolytic lysis. Rapid, complete platelet inhibition (good P2Y_12_ saturation), high heparin activity during the procedure, and optimization of microcirculation perfusion (e.g., pharmacological no-reflow treatment) indirectly limit the early “peak” of NETosis. Periprocedural glycemia may also be important: hyperglycemia promotes ROS and lowers the threshold for NETosis; therefore, strict metabolic control (in patients with diabetes) is a rational element of the IR-mitigation package. In CABG, especially with the use of extracorporeal circulation, systemic neutrophil activation occurs (contact of blood with artificial surfaces, complement), which promotes NETosis; “Off-pump” techniques, biocompatible circuits, heparinization, and shortened extracorporeal circulation time may mitigate this phenomenon. Heparin, in addition to its anticoagulant effect, binds histones and partially neutralizes their endothelial cytotoxicity, which may be important for early protection of the vascular barrier during procedures [47,149,150,151,152,153,154].

Clinically, the kinetics of NET biomarkers measured serially are useful: a baseline sample (before reperfusion), followed by a window of 0–3 h (peak IR), 24–48 h (acute-phase resolution), 7–14 days (resolution), and 3–4 months (stabilization). The panel may include cfDNA, MPO–DNA complexes, CitH3, and/or NE activity. Changes in these markers should be interpreted in parallel with markers of muscle damage (troponins), perfusion parameters (TIMI/Myocardial Blush, IMR), microcirculation characteristics on cardiac resonance (MVO/LGE), and measures of platelet activation. Such multidimensional monitoring facilitates risk stratification for IR events (no-reflow, distal embolization), selection of adjuvant therapy, and assessment of intervention efficacy [155,156,157,158].

At the level of IR-burst mitigation, three layers can be distinguished: (1) preprocedural—complete, rapid platelet inhibition (in STEMI, consider rapid-onset medications), optimization of glycemia, oxygenation, and acid-base balance; (2) intraprocedural—adequate heparin anticoagulation, careful manipulation of the thrombus (to limit distal embolization), no-reflow treatment (e.g., microcirculation vasodilators), appropriate selection of stenting strategy; (3) postprocedural—continuation of intensive antiplatelet therapy, consideration of upstream intervention on the inflammatory–NET axis (e.g., colchicine in appropriate indications), metabolic control, and—as part of studies—targeted midstream/downstream approaches (histone neutralization, NE/MPO inhibition, enhancement of network clearance). From a translational perspective, combined strategies (e.g., intensive platelet modification + NET clearance support) are of interest, as they simultaneously reduce the number of new bursts and shorten the duration of tissue exposure to toxic network components (Table 4). The practical implications remain unchanged: in PCI/CABG, it is worthwhile to schedule serial NET biomarker measurements (cfDNA, MPO–DNA, CitH3) before, immediately after, and in the following days after the intervention, as such a profile helps predict and recognize reperfusion complications, tailor supportive care, and monitor anti-inflammatory effects in the long term [148,149,150,151,152,153,154,155]. Incorporating these indicators into existing assessment pathways (biochemistry, imaging, functional parameters) creates a coherent, mechanistically grounded system of periprocedural care, in which NETs become a measurable target and marker of therapeutic efficacy.

## 6. Conclusions

This study underscores the emerging role of NETs as central mediators of immuno-inflammatory processes in the pathogenesis of CVDs. Increasing evidence indicates that CVDs should be regarded not solely as metabolic or hemodynamic entities, but as disorders fundamentally driven by dysregulated interactions between innate immunity, vascular inflammation, and thrombosis. NETs promote key pathogenic mechanisms, including LDL oxidation, ROS overproduction, and the release of macrophage-derived cytokines, thereby accelerating the initiation and progression of atherosclerotic lesions. Concomitantly, NETs exert potent prothrombotic effects by compromising endothelial integrity, inducing platelet activation and aggregation, and attenuating endogenous anticoagulant pathways.

By simultaneously amplifying atherogenic and thrombotic cascades, NETs contribute to the full spectrum of CVDs, encompassing CAD, carotid and peripheral artery disease, and cerebrovascular manifestations, as well as systemic thrombo-inflammatory complications. Their consistent detection within vascular plaques and arterial thrombi highlights their role as structural and functional determinants of plaque destabilization and acute ischemic events. From a translational perspective, targeting NET formation and persistence represents a promising therapeutic avenue. Pharmacological interventions aimed at inhibiting neutrophil activation, mitigating oxidative stress, or disrupting platelet–NET interactions, combined with interventional procedures that restore perfusion, may help attenuate both vascular injury and inflammation. Nevertheless, clinical implementation is currently limited by the absence of standardized assays, reproducible biomarkers, and validated endpoints.

In summary, converging evidence positions NETs as a mechanistic bridge linking lipid dysregulation, chronic vascular inflammation, and thrombosis in CVDs. Given the global burden of these diseases as the leading cause of morbidity and mortality, further research elucidating NET biology and developing targeted diagnostics and therapeutics is imperative to refine risk stratification and improve clinical outcomes.

## Figures and Tables

**Figure 1 cells-14-01562-f001:**
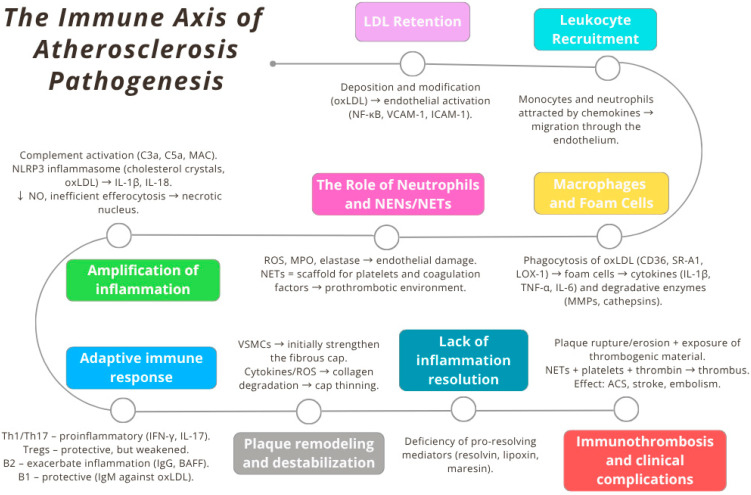
The immune axis of atherosclerosis pathogenesis (based on [19,20,21,22,23,24,25,26,27,28]). The schematic summarizes the sequential and overlapping processes that drive atherogenesis and its thrombo-inflammatory complications. Initial LDL retention and leukocyte recruitment trigger endothelial activation, leading to the formation of macrophages and foam cells. Neutrophils, through the release of NENs/NETs, amplify inflammation, promote oxidative and proteolytic vascular damage, and provide a prothrombotic scaffold. Adaptive immune responses and defective resolution of inflammation contribute to plaque remodeling, destabilization, and necrotic core formation. Ultimately, failure to restore vascular homeostasis results in plaque rupture or erosion, exposure of thrombogenic material, and the onset of immunothrombosis, leading to clinical events such as ACS, stroke, or peripheral embolism.

**Figure 2 cells-14-01562-f002:**
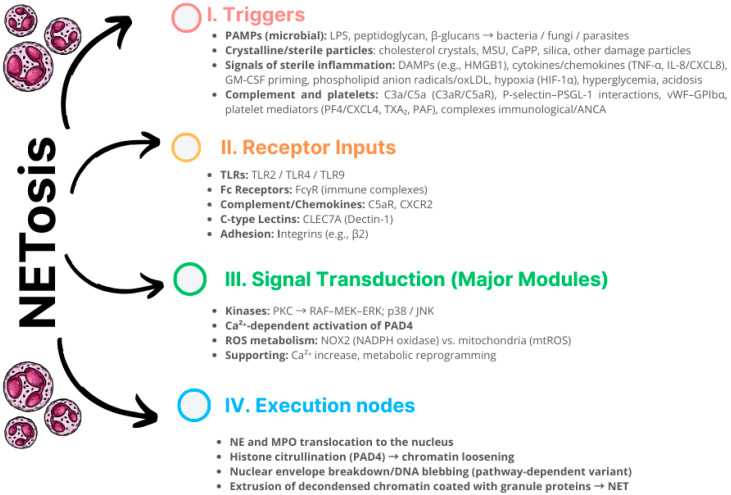
Graphical representation of the factors and processes occurring during NETosis. (I) Triggers: PAMPs, sterile particles (e.g., cholesterol crystals/MSU), DAMPs/cytokines, oxidative stress, and complement and platelet signals. (II) Receptor inputs: TLR2/4/9, FcγR, C5aR/CXCR2, C-type lectins (CLEC7A/Dectin-1), and integrins. (III) Signal transduction: PKC→RAF–MEK–ERK and p38/JNK kinase cascades; Ca^2+^ increase and PAD4 activation; ROS production by NOX2 and/or mitochondria, with concomitant metabolic reprogramming. (IV) Execution nodes: NE/MPO translocation to the nucleus, histone citrullination by PAD4, nuclear envelope breakdown/DNA blebbing, and finally, ejection of decondensed chromatin coated with granule proteins—NETs [32,33,34,35,36,37,38].

**Figure 3 cells-14-01562-f003:**
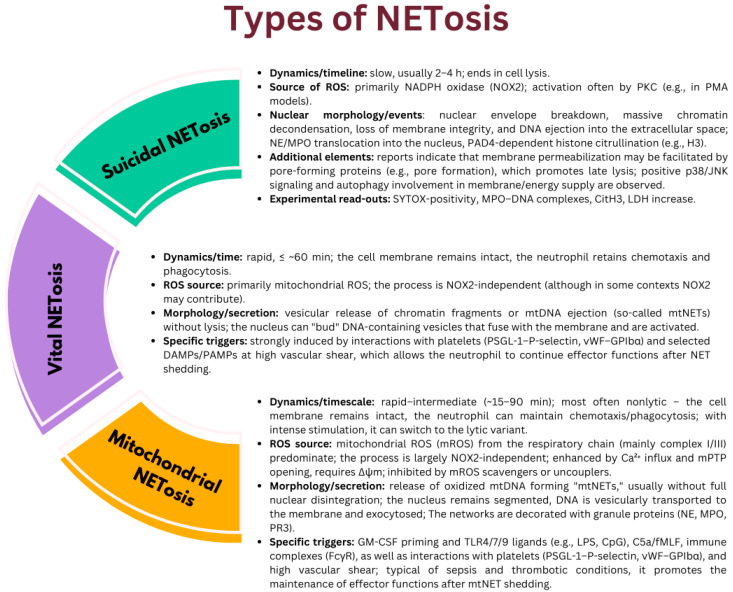
Comparison of suicidal, viable, and mitochondrial NETosis variants [39,40,41,42,43,44,45].

**Figure 4 cells-14-01562-f004:**
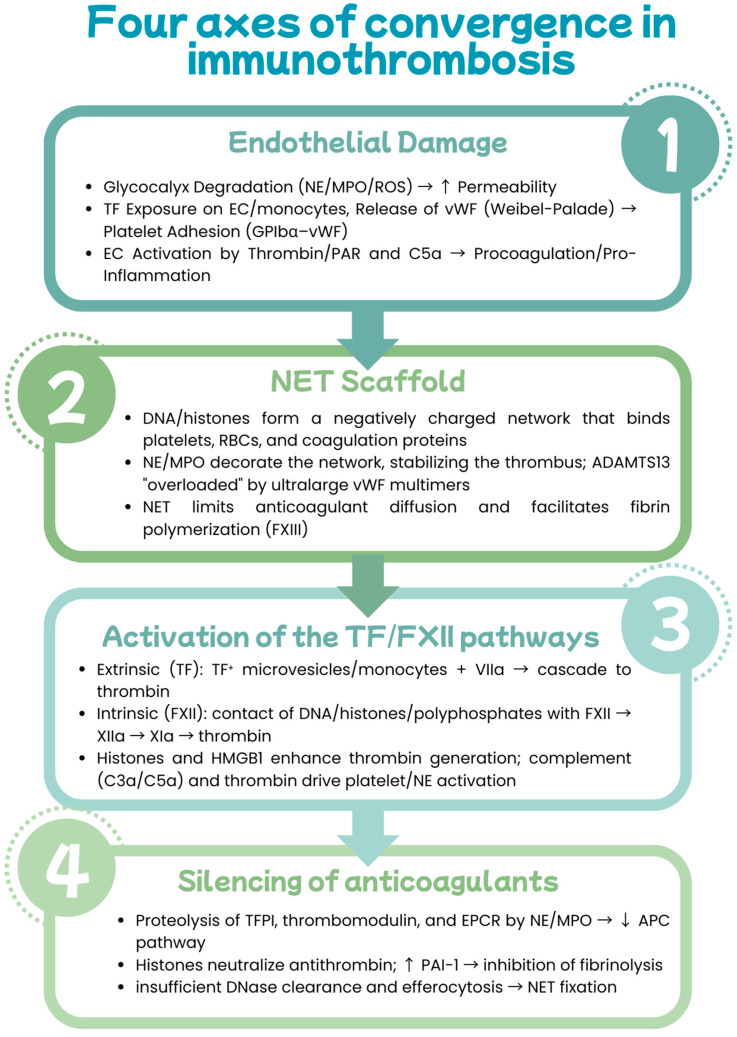
Four axes of convergence in NET-driven immunothrombosis [72,73,74,75,76,77,78,79,80,81,82,83,84,85,86,87,88,89,90,91,92].

**Figure 5 cells-14-01562-f005:**
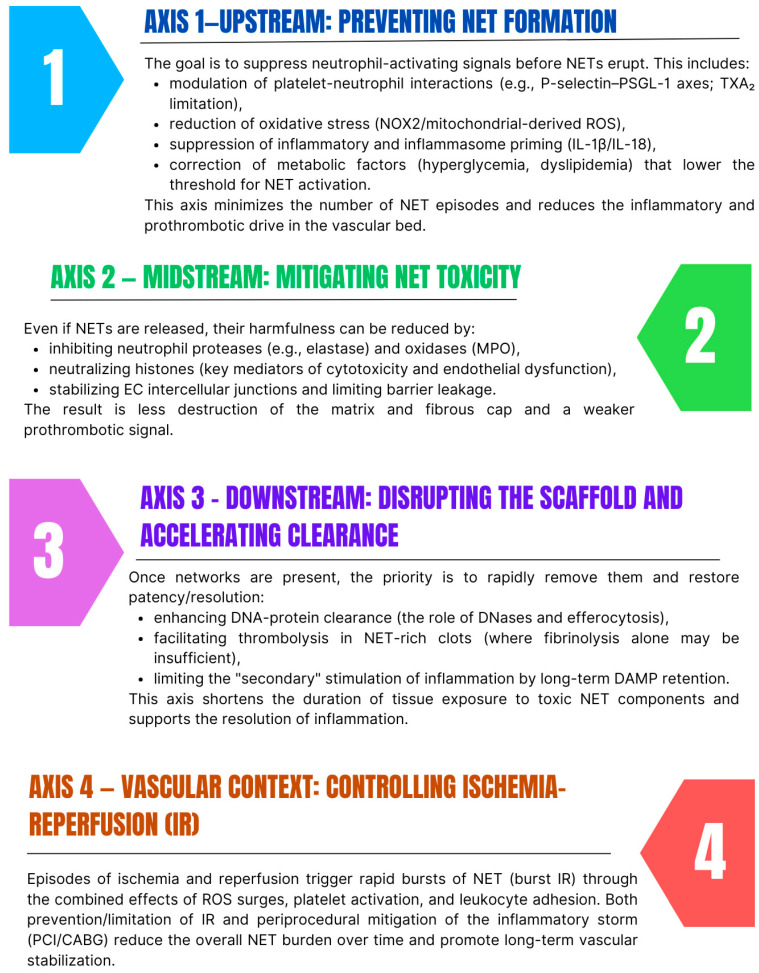
Four axes of NET control in cardiovascular diseases [own work].

**Table 1 cells-14-01562-t001:** Similarities and differences between NETs and NENs (based on [31,32,53]).

Feature/Characteristic	NETs (Classic Neutrophil Extracellular Traps)	NENs (Broader Definition of Neutrophil Extracellular Networks)
Basic Material	Decondensed nuclear chromatin (DNA, histones H2A/H2B/H3/H4) and granule proteins (MPO, NE, cathepsin G, LL-37)	Similar molecular composition (DNA, histones, proteolytic and antimicrobial proteins), but this concept also encompasses potentially diverse network variants with distinct architecture and composition
Release mechanism	NETosis in its classic variants: suicidal, vital, and mitochondrial, involving ROS, PAD4, and cell membrane disintegration.	The term encompasses both classic NET forms and other, network-like extracellular structures of neutrophils, the formation of which does not necessarily correspond to the full mechanisms of NETosis.
Immunological consequences	High pro-inflammatory properties, generation of immuno-inflammatory signals, presentation of autoantigens, initiation and maintenance of thrombosis	Similar immunological effects, but the network connection with immune cells and matrix elements is emphasized, indicating the integrative nature of the inflammatory response.
Role in pathophysiology	Well-documented role in autoimmune diseases, cancer, thrombotic processes, and in the pathogenesis of cardiovascular diseases	Conceptually useful for describing the spatial organization and multifunctionality of neutrophil structures, but less clearly characterized mechanistically
Removal/Degradation	Degradation by DNases (DNase I, DNase1L3), phagocytosis of fragments by macrophages, and other cellular “clearing” mechanisms.	The degradation mechanisms are analogous, but in the context of more extensive networks, removal failure (defect in NEN clearing) may potentially be more important.
Research and clinical applications	Used as a source of biomarkers: cfDNA, MPO–DNA complexes, citrullinated histones (CitH3), neutrophil elastase (NE)	Potential for expanding analyses to a holistic approach, encompassing network organization as an integral element of immunopathology and new translational strategies

**Table 2 cells-14-01562-t002:** NETs in selected pathophysiological conditions and cardiovascular diseases.

Unit/State	Main Mechanisms Related to NET	Biomarkers/Laboratory Changes	Vascular/Clinical Consequences	References
Arterial hypertension	EC activation and impairment of vascular function by histones; VSMC proliferation (Akt/CDKN1b/TK1); enhancement of neutropenia by Ang II, isolevuglandins	↓ pressure and better relaxation	Increased peripheral resistance; enhancement of the inflammatory–prothrombotic axis	[94,95,96,97]
Dyslipidemia	↑ NET creation by oxLDL/OxPL; ↓ clearance (DNase); IL-1β from NLRP3 induces netosis; HDL inhibits netosis	↑ NET accumulation; pro-inflammatory profile (IL-1β)	Perpetuation of intima inflammation; facilitation of LDL modification and plaque progression	[59,98,99]
Exposure to tobacco smoke	ROS-dependent NETosis; cGAS/TLR9 → NF-KB; neutrophil reprogramming (ferroptosis, “vital” NETosis); ↓DNase	↑ NET and pro-inflammatory mediators; ↓ DNase activity	Chronic inflammation, endothelial dysfunction, and increased susceptibility to immunothrombosis	[100,101,102,103]
CAD	NETs in plaques and thrombosis; antithrombolytic properties; potential induction of cardiomyocyte autophagy/apoptosis	High NET content in thrombi; presence of NE	Worse prognosis with high NET burden; difficult clot lysis; myocardial damage	[104,105,106,107]
Carotid artery disease	NETs stimulate angiogenesis, shifting the coagulation/thrombolysis balance toward coagulation.	—	Plaque destabilization (neovascularization); increased risk of thrombotic events	[108,109,110]
PAD	Atherogenic and prothrombotic effects of NETs; presence of NETs in thrombotic material	↑ MPO, CitH3, cfDNA; DNA in clots	Increasing the severity of PAD promotes thrombosis in the micro- and macrocirculation.	[111,112,113]

Abbreviations: NET/NETosis—neutrophil extracellular trap formation; EC—endothelial cells; VSMC—vascular smooth muscle cells; AKT (Akt)—serine/threonine protein kinase B; CDKN1B (p27^Kip1)—cyclin-dependent kinase inhibitor 1B; TK1—thymidine kinase 1; Ang II—angiotensin II; oxLDL—oxidized low-density lipoprotein; OxPL—oxidized phospholipids; DNase—deoxyribonuclease; IL-1β—interleukin-1 beta; NLRP3—NLR family pyrin domain containing 3 (inflammasome); HDL—high-density lipoprotein; ROS—reactive oxygen species; cGAS—cyclic GMP–AMP synthase; TLR9—Toll-like receptor 9; NF-κB (NF-kappa B)—nuclear factor kappa-light-chain-enhancer of activated B cells; CAD—coronary artery disease; NE—neutrophil elastase; PAD—peripheral artery disease; MPO—myeloperoxidase; CitH3—citrullinated histone H3; cfDNA—cell-free DNA; IsoLGs—isolevuglandins.

**Table 3 cells-14-01562-t003:** Drug classes and the NET axis—mechanisms, expected effect, and safety.

Class/Drug	NET Axis	Primary Grip Point/Mechanism	Expected Effect on NET	Key Risks/Considerations	References
**Colchicine**	Upstream	Microtubule stabilization; ↓ inflammasome (IL-1β/IL-18); ↓ neutrophil recruitment	↓ frequency of episodes of netosis	Gastrointestinal intolerance	[120]
**Aspirin**	Upstream	↓ COX (mainly COX-1) →↓ TXA_2_; attenuation of the P-selectin–PSGL-1 axis and platelet signals (PF4)	↓ platelet–neutrophil interaction→ ↓ NET	Bleeding (esp. long-term/focused therapy)	[121,122,123]
**P2Y_12_ inhibitors** (ticagrelor/prasugrel/clopidogrel)	Upstream	Block P2Y_12_ → ↓ platelet activation; (ticagrelor: effect on the adenosine system)	↓ NET triggering signals from the plates	Bleeding; differences between molecules	[124]
**Metformin**	Upstream	AMPK → ↓ ROS (mitochondria); ↓ pro-inflammatory cytokines; ↓ glycemia	↓ ROS-dependent neutropenia and hyperglycemia	Gastrointestinal complaints; rarely lactic acidosis	[125,126,127,128,129,130,131]
**Statins**	Upstream	↓ LDL/oxLDL; pleiotropy: antioxidation, NO improvement	↓ DAMP stimuli; indirectly ↓ NET	Myopathy; ↑ liver enzymes (rare)	[132,133]
**PCSK9 inhibitors**	Upstream	Deep reduction in LDL/Lp(a) → ↓ oxPL/DAMP	Potentially ↓ NET (indirectly)	Cost; injections	[134,135]
**Dexamethasone**	Upstream	Inhibition of TNF-α/IL-6; enhancement of anti-inflammatory signals	↓ NETosis in models	Immunosuppression (infections, sepsis)	[136,137,138]
**PAD4 inhibitors** (e.g., cl-amidine)	Upstream/Midstream	Block histone citrullination → inhibition of chromatin decondensation	↓ The formation of NET	No clinical data on CVD	[139,140]
**NE/MPO inhibitors** (e.g., sivelestat/MPO candidate)	Midstream	↓ cytotoxicity of NET-related proteases/oxidases	↓ EC/matrix damage; ↓ prothrombotic	Limited CVD data; particle safety	[13,32,141]
**Heparin/heparinoids**	Midstream	Histone binding; neutralization of toxicity	↓ endothelial damage; partially “anti-NET”	Bleeding; HIT (heparin)	[47]
**DNase I (dornase alfa) ± tPA**	Downstream	Degradation of NET DNA-scaffolds; synergism with fibrinolysis	↑ NET clearance; ↑ NET-rich clot lysis	No routine vascular use	[142,143]
**N-acetylcysteine (NAC)**	Upstream/Downstream	Antioxidant; disruption of NET protein disulfide bonds	↓ ROS-NET; easier clearance	Good tolerability; side effects rare	[144,145,146]

Abbreviations: NET—neutrophil extracellular trap(s); P2Y12—P2Y12 adenosine diphosphate receptor; COX (COX-1)—cyclooxygenase (cyclooxygenase-1); TXA_2_—thromboxane A_2_; P-selectin—platelet selectin; PSGL-1-P-selectin glycoprotein ligand-1; PF4- platelet factor 4; AMPK -AMP-activated protein kinase; ROS—reactive oxygen species; LDL—low-density lipoprotein; oxLDL—oxidized LDL; NO—nitric oxide; DAMP(s) -damage-associated molecular pattern(s); PCSK9—proprotein convertase subtilisin/kexin type 9; Lp(a)—lipoprotein(a); oxPL—oxidized phospholipids; TNF-α -tumor necrosis factor alpha; IL-1β/IL-18—interleukin-1 beta/interleukin-18; CVD—cardiovascular disease; PAD4—peptidyl arginine deiminase 4; NE -neutrophil elastase; MPO—myeloperoxidase; EC—endothelial cells; DNase I (dornase alfa)—deoxyribonuclease I; tPA -tissue plasminogen activator; NAC—N-acetylcysteine.

**Table 4 cells-14-01562-t004:** Periprocedural interventions and the NET axis in ischemia–reperfusion [148,149,150,151,152,153,154,155,156,157,158].

Treatment Phase	Intervention (Examples)	Effect on Neutrophil/Platelet Activation	Effect on NET Dynamics	Expected Clinical Signal	Notes/Limitations
**Before** (PCI/CABG)	Rapid platelet inhibition (full P2Y_12_ saturation); statin loading dose; glycemia, oxygenation control	↓ initial activation of neutrophils/platelets → smaller IR-burst	cfDNA, MPO–DNA, CitH3, NE; troponins	Lower risk of no-reflow/distal embolization	Risk of bleeding with intense antiaggregation
**Before (selected cases)**	Colchicine (if compatible with CVD indications)	↓ inflammatory priming → ↓ NET releases	jcfDNA, MPO–DNA, CitH3, NE; troponins and CRP/IL-6 (inflammatory background)	Potentially milder IR peak	Individual qualification: GI tolerance
**Intraprocedural PCI**	Adequate heparinization; gentle work on the clot, minimizing embolization; no-reflow treatment (microvasodilators)	↓ platelet activation and neutrophil adhesion; EC protection	ACT/anti-Xa; perfusion parameters (TIMI/MBG/IMR)	Better microcirculation perfusion	Heparin: + histone binding effect (partial neutralization)
**Intraprocedural CABG**	Biocompatible circuits; shortening CPB time; considering off-pump	↓ generalized neutrophil/complement activation	hemodynamics; hemolysis/complement markers	Smaller NET system burst	Requires experience of the center/team
**After treatment (0–48 h)**	Continue DAPT; maintain NO/endothelial protection; tight glycemic control	Suppression of secondary neutropenia stimuli; protection of the EC barrier	cfDNA, MPO–DNA, CitH3 at 0–3 h and 24–48 h; troponins	IR peak declines; perfusion stabilizes	Bleeding risk balance
**After treatment (7–14 days)**	Supporting clearance: circulatory rehabilitation, lipid optimization, NO; (research: DNase I, histone neutralization, NE/MPO inhibitors)	↑ network breakdown/clearance; inflammation quenching	cfDNA/MPO–DNA/CitH3/NE	Decrease in NET markers (resolution)	Mainly translational approaches
**Long term (1–4 months)**	Lipid optimization (statins/PCSK9), glycemic control; lifestyle modification	↓ DAMP/ROS stimuli → stable NET reduction	periodically: NET panel + lipid profile/glycemia	Sustained decrease in NET; plaque stabilization	The continuing decline in NET

Abbreviations: PCI—percutaneous coronary intervention; CABG—coronary artery bypass grafting; P2Y12—P2Y12 ADP receptor; IR—ischemia–reperfusion; cfDNA/jcfDNA—cell-free DNA; MPO–DNA—myeloperoxidase–DNA complexes; CitH3—citrullinated histone H3; NE—neutrophil elastase; EC—endothelial cells; ACT—activated clotting time; anti-Xa—anti-factor Xa activity; TIMI—Thrombolysis In Myocardial Infarction (flow grade); MBG—myocardial blush grade; IMR—index of microcirculatory resistance; CPB—cardiopulmonary bypass; DAPT—dual antiplatelet therapy; NO—nitric oxide; DNase I—deoxyribonuclease I; MPO—myeloperoxidase; CVD—cardiovascular disease; CRP—C-reactive protein; IL-6—interleukin-6; PCSK9—proprotein convertase subtilisin/kexin type 9; DAMP—damage-associated molecular patterns; ROS—reactive oxygen species.

## Data Availability

No new data were created or analyzed in this narrative review. Data sharing is not applicable. All information is derived from published studies cited in the References.

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
