# Peer review of "The Role of Neutrophil Extracellular Networks in Cardiovascular Pathology"

_cells, 2025, doi:10.3390/cells14191562_

Round 1

Reviewer 1 Report

Comments and Suggestions for Authors

Comments

Dear authors here are my suggestions:

Page 2, line 47: “Among CVDsyou must have uniformity in the text. In the previous lines you referred to CVD. CVD or CVDs?

Page 2, line 50: “The atherosclerosis paradigm now extends….” explain please the word “now”? What offers that word?

Paragr. At least a shape

Page 6, line 249: “(≤~1 hour)” please put the symbol h

The same in page 15, line 456.

Page 6, line 240: In the title “The role of NET” or “The role of NETs”? check in the whole text please

Page 8, line 318: “ [36/]” Delete please “/

Page 11, line 382: “in the vascular bed” Could you please rephrase?

Page 11, line 399: “in vivo, ….in vitro” put italic please

Page 15, line 453: “ NNETosis” delete please one “N”

Page 16, table 3: You have two columns with the same name “Intended effect on” Could please be more specific?

Page 16, table 3: “ndividual” could you please respell?

Page 21, reference 51: correct please the style of Journal

The same in references: 63, 72, 79, 144, 147, 150

Author Response

Dear Reviewer,

We would like to sincerely thank the Reviewer for the time, effort, and expertise dedicated to evaluating our manuscript entitled “The Role of Neutrophil Extracellular Networks in Cardiovascular Pathology” (Zofia SzymaÅ„ska, Antoni Staniewski, MichaÅ‚ KarpiÅ„ski, Katarzyna Zalewska, Oliwia Kalus, Zofia Gramala, Joanna Maćkowiak, Sebastian Mertowski, Krzysztof J. Filipiak, Mansur Rahnama-Hezavah, Ewelina Grywalska, Tomasz Urbanowicz). We are grateful for the constructive comments and valuable suggestions, which have enabled us to enhance both the clarity and scientific depth of our work.

Page 2, line 47: “Among CVDs you must have uniformity in the text. In the previous lines you referred to CVD. CVD or CVDs?

Thank you very much for this comment, it has been corrected in the current version of the manuscript

Page 2, line 50: “The atherosclerosis paradigm now extends….” explain please the word “now”? What offers that word?

Paragr. At least a shape

Thank you very much for this comment, it has been corrected in the current version of the manuscript

Page 6, line 249: “(≤~1 hour)” please put the symbol h

The same in page 15, line 456.

Thank you very much for this comment, it has been corrected in the current version of the manuscript

Page 6, line 240: In the title “The role of NET” or “The role of NETs”? check in the whole text please

Thank you very much for this comment, it has been corrected in the current version of the manuscript

Page 8, line 318: “ [36/]” Delete please “/

Thank you very much for this comment, it has been corrected in the current version of the manuscript

Page 11, line 382: “in the vascular bed” Could you please rephrase?

Thank you very much for this comment, it has been corrected in the current version of the manuscript

Page 11, line 399: “in vivo, ….in vitro” put italic please

Thank you very much for this comment, it has been corrected in the current version of the manuscript

Page 15, line 453: “ NNETosis” delete please one “N”

Thank you very much for this comment, it has been corrected in the current version of the manuscript

Page 16, table 3: You have two columns with the same name “Intended effect on” Could please be more specific?

Thank you very much for this comment, it has been corrected in the current version of the manuscript

Page 16, table 3: “ndividual” could you please respell?

Thank you very much for this comment, it has been corrected in the current version of the manuscript

Page 21, reference 51: correct please the style of Journal

The same in references: 63, 72, 79, 144, 147, 150

Thank you very much for this comment, it has been corrected in the current version of the manuscript

We hope that our responses will fully clarify the issues raised and that the revised manuscript will now meet the standards required for publication. Once again, we are thankful for the Reviewer’s insightful feedback, which has significantly contributed to the improvement of our work.

Reviewer 2 Report

Comments and Suggestions for Authors

General Comments

This manuscript by SzymaÅ„ska et al., entitled “The Role of Neutrophil Extracellular Networks(NENs) in Cardiovascular Pathology” is an interesting systematic review regarding the role of neutrophil activation, a process closely linked to NETosis, within which NETs production is a major phenomenon. Since this is a systematic review highlighting the pahophyisological role of NENs, some words should be spent on the background knowledge underscoring the prominent role of neutrophils both in innate immunity and its relationships with adaptive immunity, managed by lymphocytes. This is an importnat issue in that a derangement of this interaction leads to several pathophysiological states (Buonacera A. et al., Int J Mol Sci 2022), very closely related to immuno-inflammation.

Specific Comments

  1. The involvement of immuno-inflammation has been shown to be associated with carotid atherosclerosis even in older patients, as shown by the power of neutrophil to lymphocyte ratio (NLR) to stratify the disease’s severity (Corriere T., et al., Nutr Metab Cardiovasc dis 2018). This finding should be acknoledged, because underscored the possibility of monitoring the suspect, by a tool very simple to obtain in clinical practice, of the participation of immune system in the progression of atherosclerosis.
  2. Recently, Covid-19 infection, an emerging model of immuno-inflammation, provided the opportunity to successfully test the prognostic power of NLR among other routinary inflammatory biomarkers (CRP, leucocytes, and so on), in patients hospitalized for Covid-19 disease (Regolo M. et al., J Clin Med 2022), even during respiratory failure (Regolo M. et al., J Clin Med 2023), often underlied by an acute thrombophilic pattern, leading, in turn, to the derangement of cardiovascular homeostasis.

Author Response

Dear Reviewer,

We would like to sincerely thank the Reviewer for the time, effort, and expertise dedicated to evaluating our manuscript entitled “The Role of Neutrophil Extracellular Networks in Cardiovascular Pathology” (Zofia SzymaÅ„ska, Antoni Staniewski, MichaÅ‚ KarpiÅ„ski, Katarzyna Zalewska, Oliwia Kalus, Zofia Gramala, Joanna Maćkowiak, Sebastian Mertowski, Krzysztof J. Filipiak, Mansur Rahnama-Hezavah, Ewelina Grywalska, Tomasz Urbanowicz). We are grateful for the constructive comments and valuable suggestions, which have enabled us to enhance both the clarity and scientific depth of our work.

This manuscript by SzymaÅ„ska et al., entitled “The Role of Neutrophil Extracellular Networks(NENs) in Cardiovascular Pathology” is an interesting systematic review regarding the role of neutrophil activation, a process closely linked to NETosis, within which NETs production is a major phenomenon. Since this is a systematic review highlighting the pahophyisological role of NENs, some words should be spent on the background knowledge underscoring the prominent role of neutrophils both in innate immunity and its relationships with adaptive immunity, managed by lymphocytes. This is an importnat issue in that a derangement of this interaction leads to several pathophysiological states (Buonacera A. et al., Int J Mol Sci 2022), very closely related to immuno-inflammation.

Thank you very much for this comment, it has been corrected in the current version of the manuscript

Specific Comments

  1. The involvement of immuno-inflammation has been shown to be associated with carotid atherosclerosis even in older patients, as shown by the power of neutrophil to lymphocyte ratio (NLR) to stratify the disease’s severity (Corriere T., et al., Nutr Metab Cardiovasc dis 2018). This finding should be acknoledged, because underscored the possibility of monitoring the suspect, by a tool very simple to obtain in clinical practice, of the participation of immune system in the progression of atherosclerosis.

Thank you very much for this comment, it has been corrected in the current version of the manuscript

  1. Recently, Covid-19 infection, an emerging model of immuno-inflammation, provided the opportunity to successfully test the prognostic power of NLR among other routinary inflammatory biomarkers (CRP, leucocytes, and so on), in patients hospitalized for Covid-19 disease (Regolo M. et al., J Clin Med 2022), even during respiratory failure (Regolo M. et al., J Clin Med 2023), often underlied by an acute thrombophilic pattern, leading, in turn, to the derangement of cardiovascular homeostasis.

Thank you very much for this comment, it has been corrected in the current version of the manuscript

We hope that our responses will fully clarify the issues raised and that the revised manuscript will now meet the standards required for publication. Once again, we are thankful for the Reviewer’s insightful feedback, which has significantly contributed to the improvement of our work.

Reviewer 3 Report

Comments and Suggestions for Authors

I would like to begin by congratulating the authors on their manuscript.

However, before it is accepted, I would like to comment on the following:

  1. The title porperly conveys the subject of the paper.
  2. The abstract is written too mechanic and is somewhat difficult to read. Perhaps rephrasing it and better highlighting the novelty of the paper in the conclusions would be of aid. Also, several abbrevations were not explained before hand (e.g. PCI, CABG).
  3. The Introduction provides sufficient background.
  4. Section 2 is ample and could use a figure to aid in its explanations or processes.
  5. Section 3 is a proper introduction in NETs.
  6. Section 4 offers a good explanation on NET physiology. However I think the information provided is somewhat redundant and it could be useful the merge the two sections.
  7. Sections 5 and 6 are the highlight of the paper. I would recommend more focus on the clinical and practical aspects.
  8. The Conclusions mention only CAD however the title specifically mentions cardivoascular pathology so either a change to the title is in order or a rephrasing of the conclusions.

Author Response

Dear Reviewer,

We would like to sincerely thank the Reviewer for the time, effort, and expertise dedicated to evaluating our manuscript entitled “The Role of Neutrophil Extracellular Networks in Cardiovascular Pathology” (Zofia SzymaÅ„ska, Antoni Staniewski, MichaÅ‚ KarpiÅ„ski, Katarzyna Zalewska, Oliwia Kalus, Zofia Gramala, Joanna Maćkowiak, Sebastian Mertowski, Krzysztof J. Filipiak, Mansur Rahnama-Hezavah, Ewelina Grywalska, Tomasz Urbanowicz). We are grateful for the constructive comments and valuable suggestions, which have enabled us to enhance both the clarity and scientific depth of our work.

I would like to begin by congratulating the authors on their manuscript.

However, before it is accepted, I would like to comment on the following:

  1. The title porperly conveys the subject of the paper.

Thank you very much for this comment.

  1. The abstract is written too mechanic and is somewhat difficult to read. Perhaps rephrasing it and better highlighting the novelty of the paper in the conclusions would be of aid. Also, several abbrevations were not explained before hand (e.g. PCI, CABG).

Thank you very much for this comment, it has been corrected in the current version of the manuscript

  1. The Introduction provides sufficient background.

Thank you very much for this comment.

  1. Section 2 is ample and could use a figure to aid in its explanations or processes.

Thank you very much for this comment, it has been corrected in the current version of the manuscript

  1. Section 3 is a proper introduction in NETs.
  2. Section 4 offers a good explanation on NET physiology. However I think the information provided is somewhat redundant and it could be useful the merge the two sections.

We thank the Reviewer for this valuable suggestion. After carefully revising the manuscript, we decided to merge the original Sections 3 and 4 into a single, comprehensive section. The new combined section provides a concise yet coherent introduction to NETs, followed by a streamlined explanation of their physiology, thereby reducing redundancy while maintaining scientific accuracy and clarity. We believe this restructuring improves the logical flow of the manuscript and makes it more reader-friendly.

  1. Sections 5 and 6 are the highlight of the paper. I would recommend more focus on the clinical and practical aspects.

Thank you very much for this comment, it has been corrected in the current version of the manuscript

  1. The Conclusions mention only CAD however the title specifically mentions cardivoascular pathology so either a change to the title is in order or a rephrasing of the conclusions.

Thank you very much for this comment, it has been corrected in the current version of the manuscript

We hope that our responses will fully clarify the issues raised and that the revised manuscript will now meet the standards required for publication. Once again, we are thankful for the Reviewer’s insightful feedback, which has significantly contributed to the improvement of our work.

Round 2

Reviewer 2 Report

Comments and Suggestions for Authors

No further concern.

Reviewer 3 Report

Comments and Suggestions for Authors

I believe my comments have been properly addressed and that the manuscript can be published in its current form.